# Hospitals’ Collaborations Strengthen Pandemic Preparedness: Lessons Learnt from COVID-19

**DOI:** 10.3390/healthcare12030321

**Published:** 2024-01-26

**Authors:** Carine J. Sakr, Sara A. Assaf, Lina Fakih, Saada Dakroub, Diana Rahme, Umayya Musharrafieh, Beatrice Khater, Jihane Naous, Maya Romani, Joseph Tannous, Nada Zahreddine, Mohammad Fakhreddine, Mira Itani, Nader Zalaquett, Gladys Honein

**Affiliations:** 1Employee Health Unit, Department of Family Medicine, American University of Beirut, Beirut 1107 2020, Lebanon; cs56@aub.edu.lb (C.J.S.); lf32@aub.edu.lb (L.F.); sd94@aub.edu.lb (S.D.); ds07@aub.edu.lb (D.R.); 2Department of Family Medicine, American University of Beirut Medical Center, Beirut 1107 2020, Lebanon; um00@aub.edu.lb (U.M.); bk01@aub.edu.lb (B.K.); jnaous@ufl.edu (J.N.); mr39@aub.edu.lb (M.R.); 3Infection Prevention and Control Program, American University of Beirut Medical Center, Beirut 1107 2020, Lebanon; tannousj@hotmail.com (J.T.); nk13@aub.edu.lb (N.Z.); 4Faculty of Medicine, American University of Beirut, Beirut 1107 2020, Lebanon; maf54@mail.aub.edu (M.F.); mmi24@mail.aub.edu (M.I.); ngz04@mail.aub.edu (N.Z.); 5Hariri School of Nursing, American University of Beirut, Beirut 1107 2020, Lebanon; gh30@aub.edu.lb

**Keywords:** hospital preparedness, COVID-19, hospital, training, pandemic, train-the-trainer

## Abstract

Background: The COVID-19 pandemic strained healthcare systems around the world. This study aims to understand the preparedness of private remote hospitals in Lebanon to respond to the pandemic and evaluate the impact of inter-hospital collaborations on the hospitals’ readiness. Methods: A multi-centered study was conducted between August 2020 and June 2021 in ten Lebanese private remote hospitals based on a mixed-methods embedded approach where the quantitative supported the qualitative. Through the AUB-USAID (American University of Beirut-United States Agency for International Development) COVID-19 project, these hospitals received personal protective equipment and medical equipment in addition to COVID-19-related training using the Train-the-Trainer model. The quantitative part used knowledge and evaluation questionnaires and a pre–post-intervention hospital preparedness checklist. The qualitative approach adopted semi-structured interviews with a purposive sample from key hospital personnel. Quantitative data were analyzed using SPSS version 27, and a *p*-value of <0.05 was considered to be statistically significant. For the qualitative data, a thematic analysis was performed by adopting the six-phase process described by Braun and Clarke. Results: Of the 393 healthcare workers who attended the training and completed the evaluation questionnaire, 326 completed the pre- and post-training knowledge questionnaire. A significant improvement was observed in mean knowledge scores following training for infection control, nursing, and polymerase chain reaction sampling staff (*p*-value < 0.001, *p*-value < 0.001, and *p*-value = 0.006, respectively), but not for housekeeping staff. More than 93% of the participants showed high trainer and content evaluation scores. As for the hospitals’ preparedness assessments, there was a clear improvement in the pre- and post-assessment scores for each hospital, and there was a significant difference in the mean of the total scores of partner hospitals pre- and post-USAID-AUB project (*p*-value = 0.005). These findings were supported by the qualitative analysis, where nine hospitals expressed the positive impact of the USAID-AUB intervention in improving their preparedness to respond to the COVID-19 pandemic at a critical time when it was highly needed. Despite the intervention, persistent challenges remained. Conclusions: A timely and proactive collaborative program between academic/tertiary care centers and remote community hospitals that includes sharing supplies and expertise is feasible and highly effective during public health emergencies.

## 1. Introduction

Globally, throughout the COVID-19 pandemic, demands for medical care rapidly scaled up and overwhelmed the healthcare system’s capacity. A prompt and efficient response was required in terms of hospital preparedness, the training of healthcare workers, securing the needed equipment, increasing bed capacity, and managing other existing activities [1]. Studies from high-income countries like the United States and France have reported challenges in testing and caring for COVID-19 patients, securing the needed personal protective equipment (PPE), maintaining the safety of hospital staff, and expanding hospitals’ capacities to treat patients, in addition to financial concerns [1,2]. In low-income countries, the pandemic intersected with pre-existing food, security, economic, and health crises [3] and already underdeveloped health systems [4], due to lack of infrastructure and shortages in human and financial resources [5]. For example, in the lowest-income countries, there were only 113 total hospital beds per 100,000 on average, which is half the average compared to other low-income countries and 80% less than high-income ones [4]. In addition, the availability of intensive care unit (ICU) beds in low-income countries poses a significant challenge, with countries like India, Pakistan, and Bangladesh in south Asia reporting a ratio of approximately 2 ICU beds per 100,000 individuals, compared to around 33 per 100,000 in the United States and 400 per 100,000 in Europe [6]. Sub-Saharan Africa faces an even more critical situation, exemplified by Zambia, Gambia, and Uganda, with a mere 0.6, 0.4, and 0.1 ICU beds per 100,000, respectively [6]. The shortage of medical equipment further compounds these challenges, as 41 African countries collectively possessed fewer than 2000 respirators as of mid-April, with 10 nations having none at all, significantly lower compared to the United States, which had a substantial inventory of 170,000 respirators by mid-March [7]. Additionally, low-income countries grapple with a shortage of healthcare professionals, with only 0.2 physicians and 1.0 nurses per 1000 people on average, whereas high-income countries, such as the United States, boast figures of 3.0 physicians and 8.8 nurses per 1000 people [8]. Challenges to the healthcare system were particularly threatening in rural and remote areas [9]. In India, a study showed that the rural healthcare system was overwhelmed and inadequate; as a result, the pandemic turned out to be destructive to these healthcare centers [10].

In Lebanon, where this study was conducted, the pandemic coincided with the worst economic crisis in recent history [11]. The already fragile and highly privatized (85% private hospitals) healthcare system became crippled [12,13]. Compounding the issue, all private hospitals were excluded from the early response to the COVID-19 pandemic, as directed by the Ministry of Public Health (MOPH). Initially, only one public hospital was designated as the testing and treatment site, utilizing a World Bank loan for the necessary resources [14]. Subsequently, during the initial national lockdown (March–May 2020), the MOPH extended support to equip other public and private healthcare facilities across Lebanon with the essential resources, including PPE and ventilators, for treating COVID-19 patients [14]. The remaining healthcare capacities, especially ICU beds and ventilators, bearing the high influx of COVID-19 patients, were insufficient [15].

The United States Agency for International Development (USAID)-funded project “A Nation-Wide Approach to Respond to the COVID-19 Pandemic in Lebanon”, in collaboration with the American University of Beirut (AUB), came at a critical time. Through this project, AUB partnered with ten remote hospitals in Lebanon in the North, the Bekaa, and Mount Lebanon regions. This project aimed to assist these private remote hospitals in responding to the COVID-19 pandemic by providing training and online webinars on COVID-19 and delivering PPE and medical equipment to the partner hospitals.

During December 2020, just at the end of the training period, the occupancy rates of COVID-19 patients in regular and ICU units at 8 out of 10 partner hospitals in Lebanon exceeded 80%, aligning with the surge in national COVID-19 cases. The ICU bed occupancy rate for COVID-19 patients reached 83% on 8 December 2020 [16]. An additional 20,471 people tested positive for COVID-19 between 25 November and 8 December, bringing the total reported cases since 21 February 2020 to 114,658 [16]. In January 2021, following the end of the training and the first phase of PPE and equipment delivery, the country experienced the highest daily morbidities and mortalities since the onset of the pandemic, with a 32% COVID-19 positivity rate and a 90% nationwide ICU bed occupancy rate [17].

This study aims to understand the impact of the AUB-USAID initiative on the ten partnering hospitals. Specifically, it aims to evaluate the effectiveness of the Train-The-Trainer (TTT) model on the HCW’s knowledge and to examine the preparedness of these ten private remote hospitals in Lebanon in their response to the COVID-19 pandemic pre- and post-intervention and the challenges they faced, from a multi-stakeholder perspective. It is hypothesized that an increase in COVID-19-related knowledge among the trained healthcare workers from the partner hospitals and an improvement in the hospitals preparedness to respond to the pandemic will be observed following the implementation of the project.

## 2. Methods

### 2.1. Study Design and Population

This multi-centered study, conducted between August 2020 and June 2021, was based on a mixed-methods embedded approach (QUAN-qual) where the quantitative supported the qualitative [18]. The mixed-method approach is based on the triangulation technique. In this study, we used two types of triangulations: methodology and data. We used quantitative and qualitative methodologies and, eventually, we integrated the data through a narrative-weaving approach. The ten Lebanese private hospitals partnering with AUB-USAID in the COVID-19 response project were the target, and these included five hospitals in the Bekaa region, four in North Lebanon, and one in Mount Lebanon.

The quantitative part used a pre–post training evaluation and pre–post intervention hospital preparedness checklist based on the World Health Organization (WHO), Centers for Disease Control and Prevention (CDC), and European CDC recommendations on preparedness in response to the COVID-19 pandemic [19,20,21]. The qualitative approach adopted semi-structured interviews with a purposive sample from these key stakeholders.

### 2.2. Ethical Considerations

The study protocol was approved by the Institutional Review Board (IRB) at the American University of Beirut (IRB ID: SBS-2021-0005). It was conducted in accordance with the principles of the Declaration of Helsinki, and consent was obtained from each participant before the completion of the questionnaires, checklist, and interviews. Confidentiality of the respondents’ information was assured and maintained.

The findings were reported based on the Good Reporting of A Mixed Methods Study (GRAMMS) criteria [22] (see Appendix A).

### 2.3. USAID-AUB Initiative

This project aimed to assist these private remote hospitals in responding to the COVID-19 pandemic by providing training and online webinars on COVID-19, as well as delivering PPE and medical equipment to the partner hospitals. The project consisted of introductory hospital visits, pre-assessment visits, training visits, PPE delivery, and post-assessment visits. Figure 1 demonstrates the project timeline.

During each visit, the team, consisting of a physician specialized in family medicine or occupational medicine, a nurse, an infection control consultant, and a research assistant, met with key hospital staff.

#### 2.3.1. Pre-Assessment Visits

During the pre-assessment visits, the AUB team filled out a validated COVID-19 pandemic preparedness tool (Hospital Preparedness Checklist (HPC)) with input from key hospital personnel [19,20,21]. All PPE and supplies needed by the hospitals were shared with the AUB team through an Excel sheet. Finally, the team visited the emergency departments (ED), some intensive care units (ICUs), and assessed the hospital units that were recently opened to admit COVID-19 patients.

#### 2.3.2. Training Visits

Between September and November 2020, 10 training visits were conducted. A TTT model was adopted. During each hospital visit, the AUB team provided five training sessions for the hospital’s staff, mainly those in nursing, infection control, and housekeeping. These HCWs would subsequently train their colleagues. Other hospital representatives (physicians and hospital directors, etc.) also attended the sessions. The training sessions covered topics related to infection control measures, nursing care, housekeeping procedures, Polymerase Chain Reaction (PCR) sampling, and risk assessments of exposed healthcare workers. The training included interactive presentations as well as live demonstrations. Sufficient time was allowed for questions and feedback.

#### 2.3.3. PPE and Equipment Delivery

The partner hospitals received over 2.5 million pieces of PPE (face masks, impermeable gowns, gloves, N95 respirators, overalls, and surgical masks). Moreover, each hospital received one ventilator and one high-flow nasal cannula through this initiative.

#### 2.3.4. Post-Assessment Visits

Between May and June 2021, the post-assessment visits to all partner hospitals were completed. The team assessed the changes implemented throughout the project period, as well as the hospital’s progress in following the COVID-19 training and the delivery of PPE. During these visits, the team conducted one-on-one interviews with key hospital personnel (hospital director, infection control manager, nursing director, and quality director) to gain further input on the partner hospitals’ experience during the pandemic. The post-training HPC was completed during these visits as well.

### 2.4. Study Participants

Healthcare professionals from the selected private hospitals, including nurses, housekeeping representatives, physicians, and infection control personnel, who attended the designated training were enrolled. Purposive sampling was used to identify interviewees. The aim was to target four key people from each hospital: the hospital director, the infection control manager, the quality control manager, and the nursing director.

### 2.5. Data Collection

#### 2.5.1. Quantitative Data Collection

##### Knowledge and Evaluation Questionnaires

The knowledge questionnaire was designed to assess the general knowledge of COVID-19, in addition to specific questions related to each profession (Appendix A). A score of 1 was given for each correct answer, and a score of 0 was given for all other incorrect choices. The maximum scores were 5 on the infection control questionnaire, 10 on the nursing questionnaire, 4 on the PCR sampling questionnaire, and 6 on the housekeeping questionnaire.

In addition, two evaluation questionnaires were used to evaluate the content and the trainer following each training session (Appendix A). The maximum score of the content evaluation questionnaire was 24 and that of the trainer evaluation questionnaire was 20. The content evaluation score was categorized into low, medium, and high scores, with low defined as a score less than 15; medium defined as a score between 15 and 19; and high defined as a score of 20 or above. The trainer evaluation score was also categorized into low, medium, and high scores, with low defined as a score less than 10; medium defined as a score between 10 and 14; and high defined as a score of 15 or above.

##### Hospital Preparedness Checklist (HPC)

During the pre-assessment and post-assessment visits, the validated HPC was filled by the AUB team in the presence of selected hospital personnel at each partner hospital [19,21]. The objective was to assess and compare COVID-19 preparedness before and after the USAID-AUB initiative among these hospitals. The checklist had 20 sections, including structure for planning and decision making, the development of a written COVID-19 plan, elements of the COVID-19 plan, facility communication, surge capacity, consumables and durable medical equipment, human resources, the identification and management of ill patients, moving patients in the facility, patient placement, visitor access, the continuity of essential health care services, occupational health, education and training for staff, infection control and prevention, laboratory services, waste management, environmental cleaning, postmortem care, and essential support services.

Each section included different checkpoints. Two points (2) were awarded if the hospital was found to be fully complying with the checkpoint in each section (complete), one point in the case of partial compliance (in progress choice), and zero points (0) for non-compliance or non-applicable choices. There was a total of 186 checkpoints within the overall 20 sections, and therefore, the total overall achievable score was 372. The total preparedness score was categorized into 6 categories, which ranged from extremely poor to excellent (≤62 was extremely poor, 63–124; 125–186; 187–248; 249–310 was very good, and >310 was excellent) [23]. The used HPC can be found in Appendix A.

#### 2.5.2. Qualitative Data Collection

The ten partner hospital directors were contacted by the principal investigator (CJS), either through email or through a phone call, to explain the purpose of the study and to secure their approval (either orally or by email). The script in Appendix A was used by the principal investigator to contact the hospital directors.

Hospital directors who granted their approval were asked to provide the emails of their infection control manager, quality control manager, and head nurses to inform them about the study. The script in Appendix A was used by the research assistant to communicate with the participants and inform them about the study.

On the day of the interview, each participant consented using a written consent form, and all participants agreed to be quoted in anonymous form. Three interviews took place in eight hospitals, with four interviews in the others, depending on the presence of additional key personnel such as a quality control manager. A total of 32 face-to-face interviews were conducted between May and June 2021 across Lebanon. Semi-structured interviews were conducted in locations specified by the interviewees and were recorded when permitted by the participant (n = 23, 72%); otherwise, detailed notes were taken of the interview (n = 9, 28%). Arabic and English versions of the interview were prepared; however, all the interviews were conducted in English based on the participants’ preferences. Each interview lasted between 30 and 45 min.

The interview was made up of questions on the general information on the participants (gender, age, and job title), as well as main and probing questions addressing the following topics: preparedness for the COVID-19 pandemic, persistent challenges faced during the pandemic, the impact of the AUB-USAID initiative on the level of readiness, and additional support needed. The interview guide is attached in Appendix A.

The interviews were transcribed by the research team. All transcripts were anonymized using a unique identifier for each participant. The assigned code, used for citation purposes below, was composed of two letters that referred to the professional category of the participant (hospital director (HD), nursing director (NS), infection control manager (IC), and quality control manager (QL)) followed by a number that represented the name of the hospital. Then, each participant was given a number based on the chronological order of the interview. For example, participant number 13 (NS-4) codes for the nursing director of hospital number 4, and participant number 29 (HD-10) codes for the hospital director of hospital number 10.

### 2.6. Data Analysis

#### 2.6.1. Statistical Analysis

The data were entered, cleaned, and analyzed on SPSS (Statistical Package for the Social Sciences) version 27. Frequency and percentages were presented for categorical variables, whereas mean and standard deviation were presented for continuous variables. The normality of the dependent variables was tested and not met. Thus, a Wilcoxon Signed-rank test was used to compare matched pre- and post-project knowledge and HPC total and section scores. A *p*-value of <0.05 was considered to be statistically significant.

#### 2.6.2. Thematic Analysis

The 6-phase process described by Braun and Clarke was adopted in this study [24]. In the beginning, two researchers (SA and LF) independently coded two transcripts line by line. In order to prevent interpretability bias, they discussed their coding strategy and developed a theme framework for the data analysis. This allowed them to quickly find similarities and differences within the data sets. Then, the same researchers finished the open coding and began identifying new categories. The research team (SA, LF, NZ, MI, and MF) met several times to discuss the results and determine the potential themes and sub-themes. To reflect and complete the outcomes, the latter was shared with the whole team. Finally, a thorough narrative of the results was produced and backed with quotes from the individual interviews.

### 2.7. Data Synthesis

We integrated the data at the analysis level through a narrative-weaving approach, where the quantitative and qualitative findings are presented together by theme and sub-theme. The findings were further discussed for concordance, discordance, or expansion. The integration of results was assessed by the research team [25].

## 3. Results

A total of 393 healthcare professionals from the participating partner hospitals filled out the evaluation questionnaire. From these, 326 completed the pre- and post-training knowledge questionnaire. In total, 32 participants were interviewed from 10 different hospitals and they included 9 hospital directors, 10 infection control managers, 10 nursing directors, and 3 quality directors.

### 3.1. Partner Hospitals’ Characteristics

Table 1 shows the characteristics of the studied hospitals as they were defined before the intervention. Hospital 6 had the highest number of beds (n = 220), followed by hospital 2 (n = 199), with the least number of beds in hospital 7 (n = 75). The highest number of ICU beds was in Hospital 3 (n = 29). Hospital 6 also had the highest number of invasive mechanical ventilators (n = 46). All hospitals had microbiology laboratories.

In the next sections, we describe the hospitals’ preparedness and the impact of the different components of the project, specifically the capacity building through the TTT model and the provision of PPE and equipment, through the following five emerging themes: 1—measures taken to respond to the pandemic, 2—challenges faced by the hospitals, 3—the impact of the TTT on HCWs, 4—the impact of the intervention on the hospitals’ preparedness, and 5—additional support needed.

### 3.2. Theme One: Measures Taken in Response to the Pandemic Prior to the USAID-AUB Project

In response to the COVID-19 pandemic, private remote hospitals introduced several measures to improve their readiness and preparedness to host and treat the increasing numbers of COVID-19 patients. These measures included re-structuring their facilities to accommodate COVID-19 patients, introducing new operations specific to COVID-19, and improvising a network of information resources to keep up to date with the evolving situation. Yet, they were left with several challenges.

#### 3.2.1. Re-Structuring Measures

All the approached hospitals had designated COVID-19 units, with most of these units established in separate buildings. All had COVID-19 regular rooms, while eight hospitals introduced negative pressures to the ICU (Intensive Care Units).


*“In the ICU, we created a new unit specific for COVID patients with negative pressure”.*

*(IC-5)*


Most of the hospitals had to undergo renovations and introduce changes into the design of other hospitals’ units in the process of setting up the COVID-19 unit and isolating it from the rest of the units.


*“Because the architecture of the hospital isn’t much helpful and we have an open floor, we had to close with wood to separate this unit from other units”.*

*(HD-1)*


All hospitals established separate triage areas for their COVID-19 units and for any COVID-19-associated services, such as radiology and PCR testing, to separate suspected COVID-19 cases from other patients.


*“We prepared a new COVID-19 department that is separate from the hospital, even the access from the department to the old building is separate, and the radiology department is separate from the flow of other patients”.*

*(HD-9)*


Only one hospital opened an out-patient clinic to evaluate the symptoms of COVID-19 suspected cases.


*“We also opened a flu clinic”.*

*(QL-3)*


The majority of the hospitals secured PPE, oxygen supply, ventilators, and aspirators. Less than half of the hospitals acquired additional beds, laboratory kits, and technology, while a few introduced PCR-testing services.


*“Now, from April first till April 6, we still did not host any COVID-19 patients, but we opened a unit for PCR testing”.*

*(QL-10)*


#### 3.2.2. Operational Measures

All ten hospitals created proactive COVID-19 plans and policies.


*“We did a meeting, and we launched the plan with the medical director, and we decided that we must cooperate with all staff to put the plan and work on it”.*

*NS-6*


To reduce the risk of transmission, most hospitals incorporated new measures for their waste or laundry circulation in their plans. This included securing all the needed equipment and assigning specialized and trained HCWs for the proper collection, storage, transfer, treatment, and final disposal of infectious waste and contaminated laundry from COVID-19 treatment units.


*“The circulation of all waste and the personnel who evacuate the infected waste were changed and transformed specifically for COVID”.*

*HD-5*


Infection control measures specific to handling food were also introduced through the established plans by almost half of the hospitals, including the proper cleaning of kitchen utensils using routine cleaning cycles and the disinfection of patients’ food trolleys, crockery, and cutlery, as well as assigning trained HCWS for delivering, handling, and disposing of COVID-19 patients’ food.


*“Food and everything related to COVID-19 patients had a new policy in place”.*

*HD-1*


In addition, most hospitals mentioned including visitor restrictions in their plans, including limiting visits to only one visitor per patient, specifying visiting hours, and applying strict infection control measures on visitors, including temperature checking, obligatory PPE, and sterilization at the entrance.


*“All the patients and visitors were obliged to put on masks; this wasn’t found before. Even inside the hospital, we were making sure that each patient has only one person with him, and of course with a mask. At the entrance, we were asking them some questions after taking their temperature.”*

*IC-5*


Only one hospital used telemedicine to provide patient care when applicable.


*“We used telephone to manage patients with mild cases”.*

*IC-10*


Moreover, almost half of the hospitals implemented COVID-19 awareness campaigns to improve the public’s knowledge of the newly emerging virus and help them cope with the situation.


*“We also did awareness for the community around us either through the hospital pamphlets or videos that we prepared with the infection control and posted on Facebook, and I can say that we were up to the challenge.”*

*HD-1*


Interestingly, most hospitals established positive networks with the surrounding hospitals to share information about the evolving COVID-19 situation and refer COVID-19 patients when one hospital’s COVID-19 beds were fully occupied.


*“We were exporting the learning to other hospitals. Other hospitals were also coming to learn. We had multiple positive networking with other hospitals”.*

*HD-8*


Capacity building was implemented by all the partner hospitals after the USAID-AUB project, where HCWs were extensively trained on COVID-19-related topics, including patient care. Post-training assessments and continuous refreshment were also adopted by partner hospitals.


*“We are fully prepared, doctors, physicians, and nurses have been trained well and undergo continuous training. We are fully prepared”*

*QL-3*


Less than half of the hospitals mentioned that they had to recruit new staff in order to efficiently manage the influx of COVID-19 patients.


*“New employees were selected from university graduates. Old and new employees were trained to care for COVID patients”*

*IC-6*


#### 3.2.3. Informational Source

Hospitals relied on different sources of information for COVID-19-related practices and patient care, mainly the World Health Organization (WHO), the Centers for Disease Control and Prevention (CDC), and the Lebanese Ministry of Public Health (MOPH). Only one hospital adopted the Lebanese Society of Infectious Disease as its main source of information.


*“The Lebanese Society of Infectious Disease was mainly used to extract COVID-19 Info”*

*IC-4*


### 3.3. Theme Two: Challenges Faced by the Hospitals before the USAID-AUB Project

The Lebanese healthcare system was crippled by a cascade of challenges arising from the emerging pandemic and the concurrent economic crisis that was deeply hitting Lebanon. Despite all the measures taken by the collaborating hospitals, the compounded crises hindered their efforts in various areas, leading to financial difficulties, a shortage of hospital and human resources, a psychological burden on staff, and major alterations to workflow.

#### 3.3.1. Challenges in Finances and Hospital Resources

Most hospitals reported suffering from financial losses and an inability to secure the necessary materials such as PPE and oxygen, as well as an inability to repair existing equipment in the case of damage due to the extreme devaluation of the local currency and ensuing inflation, resulting in relatively expensive items found on the black market.


*“The tube for the scanner alone costs 150,000$ at the black-market rate and this is an amount that you can never secure”.*

*HD-9*


The financial crisis also hit the out-of-pocket expenditure ability of the patient population. The hospitals had to pay the bills of the uninsured and cover the difference in the billing for those insured by the government, such as the National Social Security Fund (NSSF), armed forces, and MOPH.


*“Very huge costs that are not paid from the corona patients”.*

*HD-9*


Most hospitals also mentioned infrastructure problems, such as an insufficiency of COVID-19 dedicated rooms and a lack of a dedicated laboratory for each hospital.


*“We did not even have a lab for the results we used to send them to another laboratory and wait for the results”.*

*QL-10*


Given that the financial crisis limited the ability of private pharmaceutical companies to import medications from abroad, a few hospitals also experienced medication shortages.


*“The problem of medications that we suffered to find”.*

*HD-5*


#### 3.3.2. Challenges in Workflow

A few hospitals reported changes in their normal workflow, which involved reducing their patient capacity and staff struggling to adhere to hospital protocols like wearing PPE. These challenges primarily arose from financial constraints and a shortage of essential resources.


*“We weren’t able to apply all the protocols comfortably because of this crisis.”*

*IC-3*


#### 3.3.3. Challenges in Human Resources

##### Shortage in HCWs

As for human resources, more than half the hospitals suffered from personnel shortages due to increased sick leave among infected HCWs, an exodus of physicians and nurses, resignation, and relocation to better-paid organizations as a consequence of the decreased value of their original salaries at the hospitals.


*“International organizations came and paid them in US dollars, so they left us”*

*NS-6*


##### Psychological Burden on HCWs

In addition to the financial burden experienced by HCWs, three hospitals experienced psychological hardship among their staff brought on by the pandemic due to COVID-19-related stress, heavy workloads, and fear of infection.


*“It was a hard phase emotionally, psychologically, financially and everything”*

*NS-3*


### 3.4. Theme Three: Impact of the TTT on HCWs

#### 3.4.1. Feedback on the Training Sessions

All hospitals’ participants agreed that these sessions were very helpful in terms of training newly recruited staff, updating the COVID-19-related knowledge and practices of senior staff, and providing the trained hospital representatives with the basis and materials to train other HCWs in the hospital by adopting a train-the-trainer model.


*“We all attended the trainings, and it was really helpful. Then we repeated the training to all the staff based on what we were trained by [Name of institution]”.*

*NS-4*


#### 3.4.2. COVID-19 Pre/Post-Training Results

Table 2 compares the pre- and post-training mean knowledge scores of the four training sessions. The results, concurring with the feedback from stakeholders, showed a significant improvement in the mean scores following the implemented training sessions for infection control, nursing, and PCR sampling staff (*p*-value < 0.001, *p*-value < 0.001, and *p*-value = 0.006 respectively), but not for the housekeeping session.

#### 3.4.3. COVID-19 Training Evaluation

The results showed that 96.7% and 93.1% of the participants provided high scores for the trainer and the content, respectively. Moreover, only 2.3% and 2.05% provided low scores for the trainer and the training content, respectively.

### 3.5. Theme Four: Impact of the USAID-AUB Intervention on Hospitals’ Preparedness

#### 3.5.1. Key Personnel’s Feedback

The USAID-AUB intervention was then implemented to assist the hospitals in improving their preparedness to respond to the COVID-19 pandemic and overcoming some of the hurdles. All partner hospitals acknowledged the receipt of large quantities of PPE and participation in multiple training sessions covering COVID-19-related topics through this intervention. In addition, all hospitals mentioned being assessed for pandemic preparedness using the HPC before and after the interventions (PPE delivery and training).

Overall, nine hospitals provided positive feedback on the intervention, expressing gratitude and appreciation for receiving this equipment and training at a critical time when it was highly needed.


*“Of course, this initiative came exactly when we needed it. Many of the supplies we needed were not found in the market at the time.”*

*IC-7*


Only one hospital expressed dissatisfaction with the intervention, stating that the support received did not meet their expectations in terms of equipment.


*“To be honest and we were expecting more help from the USAID than just the regular PPEs that we are using on a daily basis.”*

*HD-9*


#### 3.5.2. HPC Results: Pre- and Post-Intervention

These findings were concurred in the quantitative results. As shown in Figure 2, there was a clear improvement in the pre- and post-assessment scores for each hospital. Four hospitals (40%) achieved an excellent score (>310), five hospitals (50.0%) achieved a very good score (>248 and ≤310), and one hospital (10.0%) achieved a good score (>186 and ≤248) before the implementation of the USAID-AUB project. Following the project’s completion, eight hospitals (80.0%) achieved an excellent score and only two hospitals (20.0%) achieved a very good score.

Table 3 compares the total mean scores before and after the implementation of the project. There was a significant difference in the mean of the total scores of the partner hospitals pre- and post-USAID-AUB project (*p*-value = 0.005).

### 3.6. Theme Five: Additional Support Needed

Despite the USAID-AUB intervention being effective and timely, the challenges brought on by the ongoing economic crisis and the rise of COVID-19 cases continued to threaten the healthcare system. These constraints severely impacted the partner hospitals, who reported a critical need for support in a variety of areas, including finances and hospital and human resources.

#### 3.6.1. Financial Support

Nearly every hospital highlighted the need for financial resources to sustain themselves and remain operational.


*“We mainly need financial support of course or anything that can be provided today will help us”*

*HD-10*


#### 3.6.2. Material and Technical Support

All partner hospitals reported the need for additional technical and material support, including PPE, oxygen tanks, testing kits, medications, and heavy equipment such as ventilators and HEPA filters. In terms of hospital capacity, more beds and negative pressure rooms were still needed by a few hospitals.


*“We had to coordinate with other hospitals to send them our patients when we have no more room or equipment.”*

*(IC_10)*


#### 3.6.3. Human Resources Support

Two hospitals cited the urgent need to hire new employees to replace the departing ones.

Despite the training received, half of the hospitals agreed on the persistent need for staff training as refresher courses for their staff and as basic education for newly recruited HCWs.


*“Training for the new staff (intensive training) are needed”.*

*(NS-2)*


A few hospitals reported the need for psychological support provided to staff.


*“Psychological support is much needed. All Teams are tired (medical, nursing and admin)”*

*IC-6*


## 4. Discussion

In this multi-centered study, a significant improvement was observed in the mean knowledge scores following the implemented training sessions for infection control, nursing, and PCR sampling staff, but not for the housekeeping session. In addition, there was a clear improvement in the pre-and post-assessment scores for each hospital and a significant difference in the mean of the total scores of the partner hospitals pre- and post-USAID-AUB project (*p*-value = 0.005). This finding was supported by the qualitative analysis, where nine hospitals expressed the positive impact of the USAID-AUB intervention in improving the hospitals’ preparedness to respond to the COVID-19 pandemic at a critical time when it was highly needed. Despite the support received, most hospitals reported enduring persistent challenges, including financial constraints, shortages in medical resources, and human resource limitations.

In light of the COVID-19 pandemic, all partner remote hospitals undertook a series of proactive measures to enhance their capacity and readiness to receive and manage the influx of COVID-19 patients. These measures involved the reconfiguration of their facilities to accommodate COVID-19 cases, the establishment of dedicated operational protocols for COVID-19 care, and the development of an information network to stay updated on the continually evolving situation. Such measures were similar to the preparedness strategies adopted by different hospitals worldwide during the pandemic. Ranney et al. highlighted the importance of surge capacity planning and creating adaptable hospital treatment spaces [26]. Additionally, Kim et al. emphasized the critical role of infection prevention measures, including optimizing personal protective equipment (PPE) usage [27]. Furthermore, Kandel et al. underscored the significance of healthcare workforce preparedness and training [28]. While these strategies collectively underscored the hospitals’ proactive strategies for ensuring a swift and efficient response to the COVID-19 pandemic, healthcare institutions faced various challenges. These challenges encompassed financial constraints, scarcities in hospital and human resources, psychological burdens on staff, and substantial modifications to workflow.

During this challenging period, the USAID-AUB project provided crucial support to these remote hospitals at a pivotal juncture, arriving before the surge of COVID-19 cases in remote Lebanese areas [29], particularly amid numerous crises in Lebanon [13,30]. This support encompassed training, the distribution of PPE and equipment, and ongoing monitoring of the situation. The project’s effectiveness was validated through both quantitative and qualitative assessments. First, following the implementation of training sessions through the TTT model, the mean knowledge scores significantly improved for infection control, nursing, and PCR sampling staff, but not for the housekeeping staff. The absence of this significant improvement among the housekeeping staff may be attributed to their education level and the small sample size (n = 20). This finding was reported by several recent studies, showing that higher COVID-19 knowledge among HCWs was significantly associated with higher educational levels [31,32,33]. The significant improvement in the knowledge scores among the rest of the trained HCWs suggests that these sessions were effective in improving their knowledge of COVID-19 preventive measures and other COVID-19-related tasks, ensuring that these future trainers had gained the knowledge needed to subsequently train their respective teams reliably and precisely. More than 93% of participants gave high scores when evaluating the training content and the trainer’s knowledge and skills, affirming the training sessions’ high quality.

The effectiveness of this training model and the provided resources (PPE and medical equipment) was also apparent in the positive feedback provided by key hospital personnel following the interventions, as well as the significant difference in the mean of the total scores among the ten partner hospitals before and after the USAID-AUB project (*p*-value = 0.005). This finding was in line with the results of a study assessing the preparedness of 11 Alexandria University Hospitals in Egypt in response to the COVID-19 pandemic, using a similar validated HPC before and after the implementation of an awareness intervention (a curriculum with guidelines for pandemic preparedness and response activities) [34]. In this study, each hospital’s total post-intervention preparedness scores clearly increased compared to their total pre-intervention scores [34]. It is worth noting that only one hospital expressed dissatisfaction with the collaborative project during the interview. Incorporating qualitative research can introduce subjectivity, which is accentuated when followed by a quantitative analysis. In this specific case, the hospital’s director expressed dissatisfaction with the intervention, despite the quantitative data indicating an improvement in the hospital’s preparedness score. This interplay between qualitative and quantitative data revealed the director’s subjective perspective, which could potentially undermine the objectivity of the findings. It is crucial to note that this viewpoint was captured before the delivery of essential medical equipment, such as ventilators and high-flow nasal cannulas, which might introduce bias because the director was aware of the impending delivery.

In addition to the primary collaborative project between AUB and the remote hospitals, it is noteworthy that the qualitative analysis identified a significant collaborative strategy between hospitals in remote areas. This strategy, adopted by the majority of hospitals, has been reported as highly effective in improving the response to the COVID-19 pandemic and alleviating the impact of shortages in COVID-19-related resources, such as beds, ventilators, and skilled personnel [35].

Despite the support provided by the USAID-funded project, the collaborating hospitals continued to face multiple challenges. They all mentioned the financial stressors secondary to the country’s economic situation. Starting in October 2019, Lebanon suffered multiple concurrent crises. There was an unprecedented economic collapse followed a popular uprising due to political instability, and as a result, the Lebanese currency lost 98% of its pre-crisis value [36]. The Lebanese financial crisis ranks in the top three most severe global crises since the mid-nineteenth century [37]. This multifaceted crisis also contributed to an exodus of skilled workers across Lebanon, especially healthcare workers [38]. In these dire circumstances, the Port of Beirut 4 August 2020, blast devasted large swathes of the capital Beirut, counting multiple hospitals, including AUBMC [39]. Roughly 30% of nurses and 40% of physicians departed the country [38]. The collaborating hospitals suffered from this outflow of HCWs. Hospitals also emphasized the psychological stress faced by their HCWs. In addition to financial stress, some hospitals were working over capacity treating COVID-19-infected patients with unprecedented staff-to-patient ratios due to staff shortages. The combination of hard working conditions, low salaries, and understaffing led to burnout being detected in more than 80% of Lebanese healthcare workers [40].

Overall, several lessons were learned through this COVID-19 project, specifically regarding the timely response, the use of the TTT model to efficiently train HCWs, and the collaboration between hospitals.

First, the delivery of COVID-19 training and knowledge, in addition to the provision of essential medical equipment and PPE, to remote hospitals via the USAID-AUB project was exceptionally beneficial, arriving at a pivotal juncture preceding the surge in cases in remote areas of Lebanon. The allocation of these acquired resources and knowledge for the management of COVID-19 cases is of immense value, as it enables the prompt identification of cases and the implementation of effective control measures, thereby exerting a substantial influence on mitigating the virus’s spread [26,41].

Second, the use of the TTT model allows for the rapid, efficient, and scalable training of healthcare workers by developing a smaller group of expert trainers who can then cascade their knowledge to a larger workforce. Published data on the effectiveness of the TTT model during the COVID-19 pandemic were very scarce at that time. However, one recent study evaluating the use of a structured TTT curriculum for PPE and airway management education showed no difference in the change in comfort level between learners taught by the initial trainer and those taught by the subsequent trainers [42]. Another study assessed the TTT approach in an infection prevention and control training program where 130 HCWs were successfully trained in just three weeks, the training material was well-received, and the staff gained greater confidence in using PPE [43]. Following a train-the-trainers workshop on infection prevention and control during the COVID-19 pandemic in Nigeria, a study revealed a significant improvement in the knowledge of 61 key healthcare workers, leading to knowledge gain and the establishment of a framework for subsequent step-down training for all healthcare workers in the country [44].

Third, of paramount importance in a pandemic is the imperative to establish collaboration between well-resourced major medical centers and remote healthcare facilities, fostering the exchange of expertise and resources to boost the preparedness of these less advantaged hospitals. Indeed, collaborations between remote, rural, and community hospitals with tertiary care centers are not new, as research has proven that such collaborations enhance community health and help to fight health disparities [45,46]. During the COVID-19 pandemic, a few studies describing the impact of inter-hospital collaborations were published. In the USA, Arizona State’s public health department took on a new model for collaboration between the private, public, and federal health sectors in the state, allowing for greater ease in transfers, supplies, financial resources, and maintaining hospital capacity levels, which helped in ensuring the sustainability of all the hospitals [47]. UC San Diego Health System (UCSDHS), an academic medical center in San Diego, USA, implemented a novel support program in three community hospitals to improve cross-border (USA–Mexico) critical care during COVID-19 [48]. This program was found to be effective in identifying and filling gaps in equipment and supplies, promoting adherence to evidence-based practices, and improving staff confidence in caring for critically ill COVID-19 patients at each hospital [48]. Additionally, a continuous collaboration was established among four hospitals affiliated with the Department of Medicine at New York University during the pandemic [49]. This collaborative effort focused on developing successful strategies for effective communication, surge capacity, clinical guidelines, and staff well-being, thus overcoming the unique challenges faced by each hospital [49]. In Europe, special partnerships were established between public and private hospitals during the COVID-19 pandemic, with the aim of augmenting resources, strengthening the capacity of the health workforce, managing case surges, and enhancing public health initiatives [50].

### Strengths and Limitations

One of the strengths of this study is its multi-centered feature, which included ten remote hospitals dispersed in different Lebanese areas. Importantly, to our knowledge, this study is among the first to assess the preparedness of remote hospitals for the COVID-19 pandemic and the impact of the collaboration between tertiary medical centers and remote healthcare facilities in Lebanon. Additionally, using a mixed-methods approach enhances the overall validity of the results by cross-validating the positive impact of the USAID-AUB intervention qualitatively and quantitatively. Furthermore, the mixed-method design compares different types of data, provides nuanced perspectives of the research question, and addresses the limitations of the challenges of a single-study design by compensating with another [51].

However, our study has a few limitations. While the study’s objectives did not include prolonged tracking of the trained healthcare workers, one limitation arises from the inability to assess their COVID-19 knowledge over an extended period after the training. The long-term effectiveness of the TTT model was not assessed and measured, as the trained trainees were not followed up to evaluate future implemented training and its impact on potential trainees’ knowledge. Furthermore, despite the absence of major COVID-19-related training campaigns during the intervention period, it was unfeasible to fully account for the contextual influence of pre-existing campaigns on the hospitals’ preparedness. Additionally, the progress made by the hospital staff independent of the intervention could not be quantified in the absence of a control group. Finally, only private hospitals were included in the study; this decreases the generalizability of the preparedness results to Lebanese public hospitals, which have different characteristics.

## 5. Conclusions

This study assessed the preparedness of remote Lebanese hospitals for the COVID-19 pandemic and the impact of inter-hospital collaborations through the TTT model and medical resource distribution. The findings showed significant improvements in the healthcare workers’ knowledge and hospital preparedness scores following the implementation of the project. Additionally, positive feedback from interviewed key hospital personnel was received, and the support of the project in overcoming several challenges faced by the hospitals during the pandemic was emphasized, highlighting the success of the project in assisting remote hospitals to respond to the COVID-19 pandemic.

Timely and proactive hospital collaborations are key to being well-prepared for pandemics, providing less-equipped and less-experienced hospitals with the essential knowledge, resources, and fortitude they need to safeguard the health of their communities. Implementing these learned lessons can significantly elevate healthcare system preparedness, especially in low-income countries, and enhance responses to future pandemics and health crises.

## Figures and Tables

**Figure 1 healthcare-12-00321-f001:**
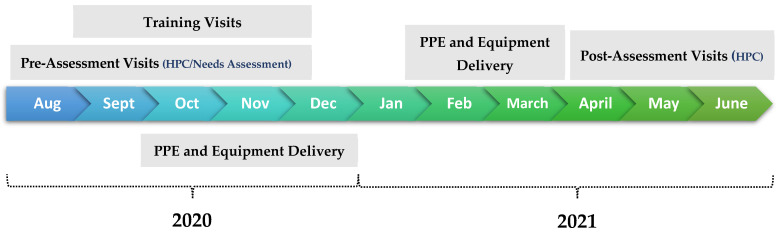
Project timeline between August 2020 and June 2021.

**Figure 2 healthcare-12-00321-f002:**
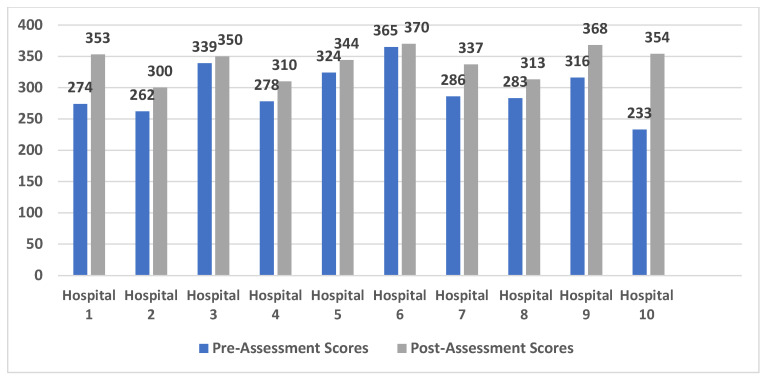
Pre- and post-assessment scores among 10 partner hospitals.

**Table 1 healthcare-12-00321-t001:** Characteristics of the partner hospitals prior to the intervention.

Partner Hospitals	Number of Beds	ICU Beds	Number of Invasive Mechanical Ventilators	Number of Non-Invasive Ventilators	Presence of Microbiology Laboratory
Hospital 1	130	7	13	5	Yes
Hospital 2	199	7	41	4	Yes
Hospital 3	125	29	20	7	Yes
Hospital 4	100	8	12	2	Yes
Hospital 5	100	21	14	12	Yes
Hospital 6	220	16	46	11	Yes
Hospital 7	75	7	18	18	Yes
Hospital 8	106	9	14	3	Yes
Hospital 9	135	28	11	11	Yes
Hospital 10	80	13	10	10	Yes

**Table 2 healthcare-12-00321-t002:** Mean scores of pre/post-training quizzes for infection control, nursing, PCR sampling, and housekeeping.

Training Session	Sample Size (N)	Pre-Training Test Mean (±SD ^#^)	Post-Training Test Mean (±SD)	Mean Difference	*p*-Value
Infection control ^1^	112	4.08 (±0.98)	4.37 (±0.86)	0.29	<0.001 *
Nursing ^2^	110	7.89 (±1.64)	8.61 (±1.75)	0.72	<0.001 *
PCR sampling ^3^	84	3.56 (±0.73)	3.75 (±0.46)	0.19	0.006 *
Housekeeping ^4,5^	20	4.95 (±0.99)	4.90 (±1.02)	−0.05	0.783

^1^ Maximum score of infection control quizzes is 5. ^2^ Maximum score on nursing quizzes is 10. ^3^ Maximum score of PCR sampling quizzes is 4. ^4^ Maximum score of housekeeping quizzes is 6. ^5^ Results are only from 4 hospitals. **^#^** Standard Deviation. * *p*-value is significant (<0.05).

**Table 3 healthcare-12-00321-t003:** Mean total score of hospital preparedness assessment checklist among 10 different hospitals before and after project implementation.

	Maximum Score	Pre-Mean Score (±SD)	Post Mean Score (±SD)	Mean Difference	95% CI		*p*-Value
					Lower Bound	Upper Bound	
Total preparedness assessment score	372	296.00 (±39.43)	339.90 (±24.50)	43.90	19.09	68.71	0.005 *

* *p*-value is significant (<0.05).

## Data Availability

The datasets used and analyzed during the current study are available from the corresponding author on reasonable request.

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
