# Peer review of "Hospitals’ Collaborations Strengthen Pandemic Preparedness: Lessons Learnt from COVID-19"

_healthcare, 2024, doi:10.3390/healthcare12030321_

Round 1
Reviewer 1 Report
Comments and Suggestions for Authors
Article presents the results of qualitative and quantitative study on hospitals' cooperations strenthening pandemics preparedness. Soma of results are quite interesting, for example comperison of qualitative to quantitative revealing director's subjective perspective. Another is assessment of TTT model.
What is missing, according to me, is the information about the epidemiology of COVID-19 in Lebanon and the study hospitals. Not detailed, but a short glance on it. Was it comparable to avarage rates, higher or lower? So, I suggest to add such information and some suggestions for potential future events in Conslusions.
Reviewer 2 Report
Comments and Suggestions for Authors
The paper of Sakr et Al. appears well presented and clear. Its usage of mixed methods approach and thematic analysis seems valid. My only concern with quantitative analysis is that it is presented as a pooled results analysis, while a multilevel analysis with nested approach would be required since it is a multicentre study, and therefore each participant is subject to higher-order level of variability. For quick reference, cfr. https://www.statisticshowto.com/nested-model-anova-factors/ or https://www.statology.org/nested-model/
Reviewer 3 Report
Comments and Suggestions for Authors
1. Introduction: Globally, throughout the COVID-19 pandemic, demands for medical care rapidly scaled up and overwhelmed the healthcare system's capacity. This applies not only to countries in Europe, but all over the world. Expecially in undeveloped countries. The introduction provides relevant data, but the underdeveloped countries thread should be expanded - it is very interesting in the context of the number of beds per 100,000 inhabitants etc.
2. You write: 'To add insult to injury, several private hospitals were not involved in responding to the COVID-19 pandemic. ' - can you explain why?
3. The study protcol is fine.
4. Figure one is not necessary - doesn't add any factual value to the text. Description in the text is enough.
5. The results section is prepared adequately, the charts are clear and easy to read.
6. In the discussion you discuss the results of the study too much and not enough reference to other studies. Please make references to similar studies in Europe, Africa.
7. Conclusion is too short - can you refer to the results?
8. References needs a little correction.
Comments on the Quality of English Language
Minor editing of English language required
Reviewer 4 Report
Comments and Suggestions for Authors
This paper is well written. however, I would like to add some comments about method section
1. Authors have used mix methods which adds value to this manuscript. however, authors need to read and add following.
2. In mix methods there are six research designs which one authors have used for this purpose read qualitative methods book written by Creswell 2009
3. Authors have to do triangulation whether data, methodology or theoretical triangulation is used need to be clarified
4. How data was transcribed. is there any interview protocol or interview guide used?
5. Add research questions in the introduction section and related them with past studies in discussion section to justify your qualitative findings.
6. Add hypotheses for quantitative findings
Comments on the Quality of English Language
Minor English spelling check is required
Round 2
Reviewer 2 Report
Comments and Suggestions for Authors
The Authors answered to all my comments, and I have no further remark.
Author Response
Comments and Suggestions for Authors:The Authors answered to all my comments, and I have no further remark.
Response: Thank you for you positive feedback.
We remain available to adjust any further comments or suggestions the editor or the reviewers might have. Looking forward to your decision regarding our manuscript.